# Valproic Acid-Induced Liver Injury: A Case-Control Study from a Prospective Pharmacovigilance Program in a Tertiary Hospital

**DOI:** 10.3390/jcm10061153

**Published:** 2021-03-10

**Authors:** Enrique S. Meseguer, Mikel U. Elizalde, Alberto M. Borobia, Elena Ramírez

**Affiliations:** Department of Clinical Pharmacology, La Paz University Hospital-IdiPAZ, Autonomous University of Madrid, 28046 Madrid, Spain; mikel-el@hotmail.com (M.U.E.); alberto.borobia@salud.madrid.org (A.M.B.)

**Keywords:** valproic acid, drug-induced liver injury, adverse drug reaction, case-control study

## Abstract

Introduction: Valproic acid (VPA) is an antiepileptic drug extensively used for treating partial and generalised seizures, acute mania and as prophylaxis for bipolar disorder. Drug-induced liver injury (DILI) persists as a significant issue related to fatal outcomes by VPA. The aim of this study was to increase our knowledge about this condition and to better identify patients affected. Methods: We conducted an observational retrospective case-control study that identified cases of DILI by VPA from the Pharmacovigilance Programme from our Laboratory Signals at La Paz University Hospital from January 2007 to December 2019. From the Therapeutic VPA Monitoring program, two control groups were assigned, VPA-tolerant patients and the other with patients who developed mild VPA-related liver injury but who did not meet the DILI criteria, matched for date, age and sex. Results: A total of 60 patients were included in the study: 15 cases of DILI, 30 VPA-tolerant controls and 15 controls with mild liver injury. Mean age for the cases was 45.7 years, 4 (26.7%) were women and 5 (33.34%) were children under 18 years, of them 3 (20%) were fatal. Polytherapy with other antiepileptic drugs (*p* = 0.047) and alcohol consumption (*p* < 0.001) were associated with a greater risk of developing DILI by VPA. A diagnosis of epileptic seizure was more frequently related to DILI when compared with the VPA-tolerant controls (*p* < 0.001). The cases developed hepatocellular liver injury (*p* < 0.001), while the mild hepatic damage controls had a higher rate of cholestatic liver injury (*p* < 0.001). The laboratory lactate dehydrogenase values were statistically higher (even at baseline) in patients with DILI than in both control groups (*p* = 0.033 and *p* = 0.039). Conclusions: VPA hepatotoxicity remains a considerable problem. This study offers interesting findings for characterising VPA-induced liver injury and at-risk patients.

## 1. Introduction

Valproic acid (VPA) is a branched short-chain carboxylic acid, established as a first-line and widely used antiepileptic agent, with a broad spectrum of activity. VPA is employed to treat partial and generalised seizures in adults and children and is considered effective against generalised tonic-clonic absences, myoclonic and partial epileptic seizures, with or without secondary generalisation. Intravenous valproate has also been shown to be useful in treating status epilepticus. In addition, it can be considered for controlling acute mania and as prophylaxis for bipolar disorder and migraine headaches [1,2].

The clinical properties of VPA were discovered in 1962 when it was tested as a solvent for other compounds (khellin derivatives) that were being investigated for potential anticonvulsant activity. The first clinical trial on epilepsy using the sodium salt of valproic acid was reported in 1964. It was introduced into clinical practice in 1967 as Depakine in France, in 1973 in the United Kingdom, and approved by the US Food and Drug Administration (FDA) in 1978 [1,3,4,5].

A broad range of mechanisms has been reported to participate in the antiepileptic and mood-stabiliser mechanisms of VPA; however, its precise mode of action has not been fully elucidated. Valproate increases the availability of synaptic gamma-aminobutyric acid (GABA) in both presynaptic and postsynaptic mechanisms. The inhibitory activity of GABA is thereby enhanced and facilitates GABA-mediated responses in specific brain regions thought to be involved in controlling seizure generation and propagation [3,6].

Other studies have shown that VPA reduces the glutamatergic N-methyl-D-aspartate (NMDA) and kainate-induced excitatory responses within the medial prefrontal cortex, suggesting that attenuation of NMDA receptor-mediated excitation is an essential mode of action for valproate’s anticonvulsant effect [7,8,9].

Valproate’s action on neuronal firing is concentration and activity dependent. This effect, critical for the anticonvulsant activity, could be explained by the direct effect on excitable membranes, which is related to the modulation of sodium, calcium, and potassium channels, especially with a use-dependent decrease in inward sodium currents. Specifically, VPA blocks both persistent and fast sodium currents [3,7]. VPA has also demonstrated an increase in extracellular levels of serotonin or 5-hydroxytryptamine (5-HT) and dopamine (DA) in the hippocampus and striatum, a change that does not appear to be related to the antiepileptic effect. In contrast, the modulation of serotoninergic and dopaminergic complexes has revealed to be relevant for the antipsychotic and neuropsychiatric actions of VPA [3,10].

Although VPA therapy has an adequate safety profile, being effective and generally well tolerated, there is a small group of patients who experience hepatotoxic reactions to this therapy, reactions that have proved to be fatal, especially in those younger than two years of age and those undergoing polytherapy [11,12]. Valproic acid hepatotoxicity has been associated with the formation of VPA-reactive metabolites, the inhibitory effect on the mitochondrial β-oxidation pathway, excessive oxidative stress and genetic variants of certain enzymes such as mitochondrial carbamoyl-phosphate synthase I (CPS1), mitochondrial enzyme polymerase gamma (POLG), glutathione S-transferase (GSTs), mitochondrial superoxide dismutase 2 (SOD2), uridine diphosphate-glucuronosyltransferase (UGTs), and cytochromes P450 (CYPs) genes. Genetic and congenital metabolic errors involving mitochondrial fatty acid oxidation, or the electron transport chain can raise the risk of VPA-induced hepatotoxicity. In particular, acute liver failure and death from valproate-induced liver injury have been reported more frequently in patients with inherited neurometabolic syndromes caused by mutations in the gene encoding POLG (e.g., Alpers–Huttenlocher syndrome). The product summary for VPA formulations warns that POLG-related disorders should be suspected in patients with a family history or symptoms that indicate a POLG-related disorder, including but not limited to idiopathic encephalopathy, refractory epilepsy (focal, myoclonic), status epilepticus as the initial clinical picture, developmental delays, psychomotor regression, axonal sensorimotor neuropathy, myopathy, cerebellar ataxia, ophthalmoplegia, and complicated migraine with occipital aura. POLG mutation testing should be performed in accordance with current clinical practice for the diagnostic evaluation of such disorders [5]. Carnitine supplementation and antioxidant administration proved to be positive treatment strategies for VPA-induced hepatotoxicity [13,14]. Despite these warnings, VPA hepatotoxicity continues to occur.

The Pharmacovigilance Programme from Laboratory Signals at a Hospital (PPLSH) is a programme based on the systematic detection of predefined abnormal laboratory values (automatic laboratory signals [ALS]), using the hospital’s laboratory information system. PPLSH has proven to be useful for the early detection and evaluation of specific serious adverse drug reactions (SADRs) associated with increased morbidity, mortality and hospital stays, as well as for gathering the detailed information necessary to study the risk factors associated with these SADRs [15].

The purpose of this study was to analyse patients who had already developed serious valproate-related hepatotoxicity detected by the PPLSH and compare them with VPA-tolerant subjects and with those patients who experienced a slight increase in transaminase levels but developed tolerance despite continuing the VPA therapy. Characterising these patients will provide more information on their clinical features, medical data and concomitant related drugs that could lead to an increased susceptibility to serious VPA hepatotoxicity and will specifically help identify those patients at risk when starting VPA treatment.

## 2. Material and Methods

### 2.1. Setting

We conducted an observational retrospective case-control study at the La Paz University Hospital in Madrid, Spain, a tertiary-care teaching facility that provides health care to a catchment area of 527,366 inhabitants in the north of Madrid. The Clinical Pharmacology Department performs all requests for therapeutic drug monitoring for the hospital and its healthcare area.

### 2.2. Definition of Cases and Controls

Cases were identified from the PPLSH of La Paz University Hospital from January 2007 to December 2019. The PPLSH is based on the proactive detection of abnormal laboratory parameters as possible indicators of SADRs. All recognised ALS are studied in order to determine their causality.

### 2.3. Case Definition

For the detection of drug-induced liver injury (DILI), we considered an increase in alanine aminotransferase (ALT) levels 5 times the upper limit of normal (×5 ULN). Based on the 2011 recommended modified biochemical criteria for identifying DILIs by the International Serious Adverse Events Consortium (iSAEC), we defined DILI when any of the following items were met: (1) ALT levels ≥5 ULN; (2) alkaline phosphatase (ALP) levels ≥2 ULN, especially in patients with elevated 5′-nucleotidase or gamma-glutamyl transferase (GGT) levels and with no bone-disease-related increase in ALP levels; and (3) ALT levels ≥3 ULN and total bilirubin (TB) levels ≥2 ULN [16,17]. The procedure for DILI detection, evaluation and classification has been described elsewhere [15]. Briefly, in phase I, on-file laboratory data at admission or during the hospital stay were screened for an ALT signal 7 days a week, 24 h per day. In phase II, the patients were identified to avoid duplicates, and electronic medical records were reviewed. In phase III, a case-by-case study was performed for the remaining cases. Considering VPA related DILI as a diagnosis of exclusion where no other explanation could be found. The cause and type of liver injury were established according to the iSAEC consortium. The causality assessment of drugs was performed using the Roussel Uclaf Causality Assessment Method (RUCAM). A causality score according to RUCAM greater than 5 (6–8 probable; ≥9 highly probable) was considered VPA-induced liver injury [18,19].

### 2.4. Control Definition

Every identified case was matched with two VPA-tolerant controls, patients who underwent VPA therapy without developing valproate-related ADRs or any kind of previous liver disease. For this purpose, we selected individuals from the Therapeutic VPA Monitoring Programme (Labtrack program, TrackHealth, Woolloomooloo, Australia) where requests for serum valproate concentrations are registered and processed by the Clinical Pharmacology Department. These controls were paired by sex and age groups (newborn, <28 days; infant, 1–23 months; preschool, 2–4 years; infancy, 5–13 years; adolescence, 14–17 years; young adult, 18–35 years; adult, 36–65 years; and older adults, >65 years). Subjects should have received VPA therapy for at least one year.

Each case was also matched with one control patient who developed mild VPA-related ALT elevation (ALT levels >1–5 ULN). Subjects were also paired by sex and group age. The only exclusion criterion was the presence of liver disease from any cause other than VPA therapy.

### 2.5. Data Collection

A total of 168 variables were collected for each case and control, including clinical variables such as family and medical history, the reason for prescribing VPA or concomitant treatment, patient characteristics (sex, age, weight, etc.), laboratory data (ALT, aspartate transaminase, bilirubin (AST), prothrombin time, etc.), and serum VPA concentration. Information on toxics, such as alcohol consumption was obtained from medical records, based on the physician’s clinical judgement in what they considered to be an excessive alcohol ingestion that could affect patient’s health.

The definition of drug interactions has been defined on the basis of “Consejo General de Colegios Oficiales de Farmacéuticos” (General Council of Official Associations of Pharmacists) of Spain. Red is used for severe interactions which must be avoided; yellow to express moderate interactions that need to be aware and treatment modification considered; green for minor interactions that should be known but not imply change of medication.

For the case report form, onset date was defined as the first day on which the patient presented any abnormal liver value (AST, ALT, ALP, or GGT). The baseline value was considered as the last value obtained prior to the onset date. The peak value was considered the maximum value reached after the onset date. The recovery value was the first laboratory value within the normal range after the onset date or the last available value.

### 2.6. Data Analysis

#### 2.6.1. Sample Size Calculation

Accepting an alpha risk of 0.05 and a beta risk of 0.2 in a bilateral contrast, 15 cases and 45 controls are required to detect a minimum odds ratio of 8A proportion of exposed subjects in the control group has been estimated to be 0.35–0.4. A follow-up loss rate of 0% has been estimated. The POISSON approximation has been used [20].

#### 2.6.2. Statistical Analysis

Frequency results are expressed in absolute terms as percentages, and the continuous variables are presented as mean (standard deviation, SD) or median (range) according to the normality test (Kolmogorov–Smirnov test). To estimate the differences between variables, a chi-squared test was employed for the categorical variables and Student´s *t*-test for the continuous parametric variables. In case of not following a normal distribution, non-parametric tests have been used (Mann–Whitney’s U or Kruskal–Wallis test, as appropriate). 

We performed a multiple logistic regression based on 1000 bootstrap samples to estimate the standard error and confidence intervals for the coefficient of determination (pseudo-R2). Data was analyzed using the statistical analysis software SPSS 20.0 (IBM Corporation, Armonk, NY, USA).

### 2.7. Ethical Statement

The study was approved by the research ethics committee of La Paz University Hospital of Madrid (PI-3970). The requirement for informed consent was waived due to retrospective data collection. The programme was conducted according to the Spanish Personal Data Protection Law [21].

## 3. Results

A total of 60 patients were included in the study: 15 cases of DILI, 30 VPA-tolerant controls and 15 controls with mild liver injury. The mean (SD) age for the cases was 45.7 (30.8) years, out of them 66.7% were adults, and only 26.7% were women. There were no statistically significant differences in age, weight, height, and body mass index (BMI) between any of the studied groups (Table 1).

In terms of family history, there was the interesting presence of an undiagnosed neurodegenerative disorder in a patient from the case group. Moreover, the presence of hydrocephalus (6.7% vs. 0%, *p* = 0.012), focal motor epilepsy (20% vs. 13.3%, *p* = 0.05), and Lennox–Gastaut syndrome (6.7% vs. 0%, *p* = 0.012) was greater in the cases than in the mild liver injury group. There was an increased presence of generalised epilepsy (26.7% vs. 40%, *p* = 0.035) and bipolar disorder (6.7% vs. 13.3%, *p* = 0.09) in the patients with mild ALT elevation than in the cases. Compared with the cases, the VPA-tolerant control group had a higher prevalence of bipolar disorder (6.7% vs. 20%, *p* = 0.003) but a lower rate of focal motor epilepsy (20% vs. 10%, *p* = 0.047) (Table 1).

The indication for VPA therapy, showed a major presence of seizures among the cases than in the VPA-tolerant controls (86.7% vs. 66.7%, *p* < 0.01), but there were no differences between the cases and the mild liver injury controls. Drug interactions were also more severe (red: 46.7% vs. 13.3% vs. 13.3, *p* < 0.01) in the medicines prescription for the patients with DILI than in the patients with mild ALT elevation and the VPA-tolerant group (Table 1). Among the possible CYP interactions, only rufinamide and phenytoin are found as inducers and clobazam as a weak inhibitor.

There were found disparities in the type of hepatitis, with hepatocellular hepatitis more common among the cases and cholestatic hepatitis more frequent among the patients who developed mild liver injury. There were no differences in the mixed type of hepatitis. The time between the start of treatment with VPA and the onset of liver injury was significantly shorter in the cases (61.6 vs. 304 days, *p* < 0.01), and the DILI duration until the recovery of the laboratory values was also shorter in the cases (18.7 vs. 204.4 days, *p* = 0.02). The hepatitis outcomes also showed statistically significant differences; as expected, there were more deaths in the cases group than in the mild liver injury control group (Table 1). 

Regarding the laboratory variables, there was a higher elevation of lactate dehydrogenase (LDH) levels in all phases (baseline, onset, peak, and recovery) in the cases compared with the mild liver injury group (Table 1). 

Finally, the multiple logistic regression (Table 2) showed that, among the toxics, alcohol achieved a statistically significant difference (*p* < 0.001; confidence interval, 6.17–8.44). The use of concomitant medication was also significant (*p* = 0.001), especially the use of concomitant antiepileptics (*p* < 0.001). There was a statistically significant difference in the major indication of VPA therapy for epilepsy (*p* = 0.002). The LDH elevation in the cases versus the control groups showed a confidence interval of 1.18–1.64 (*p* = 0.02).

## 4. Discussion

This study presents the data reported during the last 13 years from La Paz University Hospital, a total of 15 cases of VPA-induced DILI, covering all ages.

The importance of polytherapy in the development of DILI observed in the study should be considered for all patients undergoing VPA therapy. This correlation between liver damage caused by VPA and co-medication was corroborated by the chi-squared test and the multiple logistic regression and is consistent with the available literature [22,23].

There is conflicting information regarding the sex distribution of DILI reported in the literature, with a number of articles indicating a slight female predominance (58.47% and 53%) [21,23], while other studies have observed a propensity for the male sex (54%) [23]. This study found a greater incidence of liver injury in the male population (73.3%). Although another published article reported similar data (75%) [24], these observations could be explained by simple chance when employing a small sample size. If we look at series with large numbers of collected cases, the sex ratio tends to be more balanced (58.47% and 46% of female patients) [22,23].

A strong association between alcohol consumption and VPA-induced liver injury is exposed in the study. This observation has been mentioned in some published articles related with hyperammonaemia [13,25] but it is not usually described in published case reports [24,26,27], even in larges series [22,23].

The liver is known to be the main organ responsible for metabolizing ethanol, thus it is conceivable that ethanol and its metabolites can exert a direct cytotoxic effect. Hepatic metabolism of the ethanol proceeds via oxidative and non-oxidative pathways. The main steps of the oxidative pathway are mediated by alcohol dehydrogenase (ADH) and acetaldehyde dehydrogenase (ALDH) that transform ethanol to acetaldehyde and acetaldehyde to acetate, respectively. Electrons from alcohol are transferred to NAD+ by ADH. Changes in NADH/NAD+ ratio may affect biochemical reactions in the mitochondria and gene expression in nucleus. The burn of NADH requires additional oxygen amount in the mitochondria; the hepatocytes take up more than their normal share of oxygen from arterious blood but not enough to adequately supply all liver regions. Thus, alcohol consumption results in significant hypoxia of the perivenous hepatocytes that are the first ones to show evidence of damage from chronic alcohol consumption [28,29,30].

Hepatotoxicity caused by VPA is believed to be generally mediated by either an inhibitory effect of VPA on the mitochondrial β-oxidation pathway, or by VPA-induced metabolic effects, which give rise to hepatic steatosis. Glucuronic acid conjugation is the principal pathway for the metabolism of VPA in the endoplasm reticulum (ER). There is approximately 20–70% of VPA excreted in the urine as glucuronide conjugated. Β-oxidation accounts 12–40% of the administered dose of VPA in patients receiving monotherapy. In the cytosol VPA is activated to form VPA-coenzyme (CoA). Then VPA-CoA may enter the mitochondria via the “carnitine shuttle” system. In the mitochondria, VPA is then beta-oxidized, intermediates are 2-ene-VPA, 3-keto-VPA, propionyl-CoA, and pentanoyl-CoA (converted to propionyl-CoA and Ac-CoA). Within mitochondria, the CoA-derivatives are reconverted to the carnitine derivatives. In contrast to CoAs, carnitine derivatives can exit the mitochondria and the hepatocyte and be eliminated via the kidney. This is the reason for a (possible) carnitine depletion. Oxidative stress in the mitochondria is due to inhibition of the electron transport chain by VPA/metabolites, leading to the formation of superoxide. Via superoxide dismutase 2, hydrogen peroxide is produced, which can be reduced by glutathione [14]. In addition, VPA also induced hepatotoxicity by involving lysosomal membrane leakage as well as reactive oxygen species (ROS) formation which as a result of metabolic activation by CYP2E1 [11,31]. CYP2E1 is an effective enzyme for ROS production and is one of the most powerful inducers of oxidative stress in cells [11,32]. VPA-induced ROS formation seemed to be protected by inhibitors of CYP2E1. Contrarily, in vitro assays have proved the modulatory effects of ethanol, a CYPE2E1 inducer, enhancing toxicity in VPA-exposed cells [33]. The sum effect of these mechanisms, could explain the results obtained in the study.

In our study, there was no correspondence between BMI and DILI predisposition. The cases presented an in-range BMI (23.1), similar to those with mild ADR (22.7) and the VPA-tolerant controls (24.7), with no statistically significant differences. Based on these observations, the patients with abnormal BMI (underweight or overweight) should not be considered at greater risk of VPA-induced DILI. 

The liver damage duration was found to be shorter in the cases than in the mild ADR controls, which could be explained by the discontinuation of VPA treatment in all of the cases due to increased severity, while most of the control patients continued VPA therapy until tolerance was developed.

The indication for VPA therapy differed among the cases and controls. Bipolar disorder was more frequently the indication in the VPA-tolerant group, while epilepsy was more recurrent in the cases than in both control groups, presumably due to the disease’s nature, lower need for co-medication and lower dosages employed in bipolar therapy. Patients with seizures undergoing VPA treatment should therefore be closely monitored.

Time difference between the start of treatment and the diagnosis of the ADR among the cases and mild liver injury controls, could be explained by the need of the most severe patients to seek medical care whereas mild liver injury in most controls could have been found out in routine laboratory analysis.

Among the laboratory values, LDH elevation and VPA concentrations were of special interest. There was a statistically significant difference in VPA concentrations between certain measurements in each group, although this difference should not be considered clinically relevant. All of the mean VPA blood concentrations in this study were within the classical therapeutic range of 50–100 ug/mL [34,35,36], which suggests that although high VPA levels could be involved in the development of acute hepatitis, personal predisposing factors are significantly involved in liver injury, with minor effect of VPA blood concentration levels. Therefore, the identification of these factors takes on major importance.

The LDH measurements showed constantly higher levels (even at baseline) in the cases with DILI than in both control groups. Given that LDH is considered a good indicator of cellular damage and necrosis in many cell lines [37,38], this increase could be due to a greater predisposition for cell injury and destruction of any nature in patients who developed drug-induced hepatitis and might be an interesting marker for DILI susceptibility.

Referring to pharmacogenomics, numerous studies have demonstrated that specific human cytochrome P450 (CYP) enzymes has a crucial role in the metabolism of VPA. The key CYP-mediated branch of the VPA pathway produces the metabolite of 4-ene-VPA by CYP2C9, CYP2B6, and CYP2A6, which may be linked to VPA-induced liver injury. At present, the majority of reports have focus on the polymorphisms of CYP and Uridine 5′-diphospho-glucuronosyltransferase (UGT) candidate genes [11,39]. The catalysis of VPA metabolism by CYP enzymes could accelerate the formation of metabolic products such as 4-ene-VPA, thereby increasing the risk of mitochondria stress and liver toxicity [40,41]. Because of the inconsistent results about the influences of CYPs genetic variants on VPA pharmacokinetics, larger cohorts are needed to verify these results and examine the newer candidate genes [11].

In addition to pharmacokinetics, different candidate genes have been thought to induce VPA liver injury, including CPS1 and polymerase-γ-gene (POLG) mutation. Genetic mutations in POLG were confirmed to be significantly related to VPA-induced hepatotoxicity [11,42]. For this reason, VPA is contraindicated in patients with POLG variations. This POLG is defined as mitochondria DNA polymerase that is related with various disorders such as Alpers–Huttenlocher Syndrome, which is associated with an increased risk of fatal VPA liver toxicity [11]. Therefore, POLG mutation testing should be carried out in patients with suspected mitochondria disease before VPA treatment.

The study’s limitations included the small sample size used compared with other large case series registered, such as the series by Schmid et al. (2012) that collected all cases (*n* = 132) of serious VPA-related hepatic adverse effects reported to the German Federal Institute for Drugs and Medical Devices between 1993 and 2009 [22]. There is also the publication of the individual case safety reports (ICSRs) to the World Health Organisation’s global database VigiBase, which recorded 268 ICSRs related to VPA and fatal outcomes in children 17 years or younger, reported from 25 countries since 1977 to June 2013, with a total of 156 hepatotoxicity-related deaths [23]. These studies contribute to the characterisation of ADRs caused by VPA, whereas most other articles in the literature are limited to case reports. Our article could offer greater comprehension of the subject by presenting a case-control study with two different control groups.

The study’s methodological limits are our sources of case data. It is possible that the PPLSH lost some DILI cases during the data collection process. Although this number should be low, given that the alternative causes for DILI presented are well known, we cannot rule out a possible underestimation of cases. As a pharmacovigilance study, it is retrospective which could affect to the quality of the data, as it was based on previously registered information susceptible of biases. The single-centre design was another limitation; however, La Paz University Hospital is an important reference centre for pharmacovigilance in Spain.

## 5. Conclusions

Based on the findings and limitations of this observational retrospective study, certain considerations should be considered before starting VPA therapy to prevent the onset of hepatitis. In-range VPA blood concentrations and the absence of BMI-related risk factors are not sufficient determinants for preventing liver damage. Co-medication, seizure indication, known alcohol consumption, and increased baseline LDH levels are important factors that should be closely monitored to improve the safety profile of this antiepileptic drug.

### 5.1. What Is Already Known about This Subject

-Valproic acid (VPA) therapy is known to cause liver injury in a small percentage of patients, with some fatal outcomes.-Mechanisms associated with an altered mitochondrial β-oxidation pathway and excessive oxidative stress have been postulated.-Certain risk factors (such as polytherapy and younger age) could determine the chance of developing hepatotoxicity.

### 5.2. What This Study Adds

-A greater characterisation of VPA hepatotoxicity (e.g., predominance of hepatocellular vs. cholestatic hepatitis).-A number of other risk factors appear to be related to VPA-induced liver injury, such as alcohol consumption, seizures and increased baseline LDH levels.-Considering these factors, patients at higher risk of liver injury should be identified and closely monitored.-Future research is needed to confirm these findings.

## Figures and Tables

**Table 1 jcm-10-01153-t001:** Characteristics of cases Drug-induced liver injury (DILI) by valproic acid (VPA) compared with mild liver injury controls and with VPA-tolerant controls.

	Cases DILI	Mild Liver Injury Controls	Cases vs. Mild Liver Injury Controls	VPA-Tolerant Controls	Cases vs. VPA-Tolerant Controls
	(*n* = 15)	(*n* = 15)	*p* Value	(*n* = 30)	*p* Value
Age, years *	45.73 (30.8)	46.2 (31.1)	0.967	44.70 (29.8)	0.840
Adults, n (%)	10 (66.7)	10 (66.7)	1.00	20 (66.7)	1.00
Median age, years (range)	68 (27–85)	69 (19–80)	0.977	68.5 (32–77)	0.890
Children, n (%)	5 (33.39)	5 (33.39)	1.00	10 (33.3)	1.00
Median age, years (range)	9 (2–17)	9 (3–17)	0.977	9 (2–16)	0.89
Women, n (%) *	4 (26.7)	4 (26.7)	1.00	8 (26.7)	1.00
Weight, kg	58.1 (25.4)	54 (25)	0.662	58.0 (23.3)	0.580
Height, cm	153 (32)	143.7 (24.2)	0.647	155.5 (11.2)	0.18
BMI, kg/m^2^	23.1 (6.8)	22.7 (5.6)	0.918	24.7 (4.2)	0.16
Family history					
Neurodegenerative disorder without diagnosis	1 (6.7)	0	0.012	0	0.012
Medical history, *n* (%)					
CNS mass	4 (26.7)	5 (33.3)	0.28	7 (23.3)	0.623
Hydrocephalus	1 (6.7)	0	0.012	0	0.012
Dravet syndrome	1 (6.7)	1 (6.7)	1	0	0.012
Generalised epilepsy	4 (26.7)	6 (40.0)	0.035	8 (26.7)	1
Focal motor epilepsy	3 (20)	1 (6.7)	0.003	3 (10)	0.047
Bipolar disorder	1 (6.7)	2 (13.3)	0.091	6 (20)	0.003
Lennox Gastaut syndrome	1 (6.7)	0	0.012	1 (3.3)	0.31
Myoclonic epilepsy	0	0	−1	1 (3.3)	0.081
Schizoaffective disorder/schizophrenia	0	0	−1	3 (10)	0.001
Temporal seizures	0	0	−1	1 (3.3)	0.081
Previous drug allergies	1 (6.7)	4 (26.6)	<0.001	3 (10)	0.300
	Azithromycin	Amoxicillin clavulanic Itraconazole Propyphenazone Unfinished study		Metamizole (2) Amoxicillin clavulanic	
Indication of VPA therapy					
Seizures	13 (86.7)	14 (93.3)	0.111	20 (66.7)	<0.001
Bipolar disorder	2 (13.3)	1 (6.7)	0.091	10 (33.3)	<0.001
VPA dosage, mg/kg/day	26.4 (13.5)	24.5 (24.0)	0.795	21.7 (11.1)	0.213
Treatment time to ADR, days	61.6 (73.8)	304 (309)	<0.001	NA	-
Toxics, *n* (%)					
Alcohol	8 (53.3)	2 (13.3)	<0.001	1 (3.3)	<0.001
Tobacco	2 (13.3)	2 (13.3)	1	6 (20)	0.180
Alcohol and Tobacco	1 (6.7)	1 (6.7)	1	1 (3.3)	0.310
Concomitant medication	4.27 (3.1)	4.2 (3.6)	0.957	3.6 (2.8)	0.759
Concomitant antiepileptics, median (range)	2 (0–3)	1 (0–2)	0.047	2 (0–3)	0.049
Levetiracetam	4	5	-	2	-
Clobazam	2	2	-	1	-
Clonazepam	2	0	-	4	-
Lamotrigine	1	0	-	1	-
Gabapentin	2	0	-	0	-
Diazepam	1	0	-	1	-
Rufinamide	1	1	-	0	-
Phenytoin	0	0	-	2	-
Lacosamide	0	0	-	1	-
Oxcarbazepine	0	0	-	2	-
Tiagabine	0	1	-	0	-
Topiramate	0	2	-	0	-
Brivaracetam	0	1	-	0	-
Interactions ^#^					
Red	7 (46.7)	2 (13.3)	<0.001	4 (13.3)	<0.001
Yellow	9 (60)	2 (13.3)	<0.001	4 (13.3)	<0.001
Green	3 (20)	0	<0.001	0	<0.001
Liver Injury					
Hepatocellular	11 (73.3)	1 (6.7)	<0.001	NA	-
Mixed	2 (13.3)	2 (13.3)	1	NA	-
Cholestatic	1 (6.7)	8 (53.3)	<0.001	NA	-
Severity					
Acute liver injury	7 (46.6)	15 (100)	<0.001	NA	-
Severe liver dysfunction (INR > 1.5)	1 (6.7)	-		NA	-
Acute liver failure, INR > 1.5 and any degree of encephalopathy:					-
Encephalopathy G1	6 (40)	-		NA	-
Encephalopathy G2	0	-		NA	-
Encephalopathy G3	1 (6.7)	-		NA	-
Encephalopathy G4	0	-		NA	-
Liver Injury duration, days	18.7 (13.9)	204.4 (241.5)	0.02	NA	-
Outcome, *n* (%)					
Recovered	12 (80)	12 (80)	1	NA	-
Death	3 (20)	1 (6.7)	0.003	NA	-
Not recovered	0	2 (13.3)	0.012	NA	-
Laboratory Data					
Baseline					
ALT, UI/L	24 (12)	23 (18)	0.877	17 (10)	0.706
AST, UI/L	23 (8)	29 (16)	0.28	21 (10)	0.637
GGT, UI/L	55 (57)	42 (33)	0.528	32 (35)	0.041
ALP, UI/L	85 (54)	94 (38)	0.7	91 (58)	0.737
TB, mg/dL	0.6 (0.3)	0.5 (0.3)	0.537	1.7 (5.9)	0.214
PA, %	82.5 (17.5)	76.8 (33.9)	0.766	84.7 (26.3)	0.43
LDH, UI/L	333 (55)	227 (26)	0.033	202 (67)	0.039
Cr, mg/dL	0.79 (0.79)	1.0 (0.62)	0.21	0.71 (0.30)	0.108
Albumin, g/dL	4.4 (1.4)	9.5 (17.0)	0.409	3.7 (0.7)	0.444
Protein, g/dL	5.7 (2.0)	6.8 (0.9)	0.922	11.2 (16.1)	0.121
Ammonium, µmol/L	58 (-)	-	-	-	-
Onset					
ALT, UI/L	435 (347)	200 (387)	0.025	NA	-
AST, UI/L	446 (333)	163 (405)	0.009	NA	-
GGT, UI/L	207 (95)	77 (87)	0.042	NA	-
ALP, UI/L	166 (193)	147 (103)	0.824	NA	-
TB, mg/dL	1.6 (1.8)	0.6 (0.2)	0.036	NA	-
PA, %	74.4 (28.7)	85.3 (16.0)	0.37	NA	-
LDH, UI/L	580 (36)	428 (35)	0.009	NA	-
Cr, mg/dL	0.82 (0.48)	0.75 (0.54)	0.668	NA	-
Albumin, g/dL	3.4 (0.6)	2.8 (1.5)	0.374	NA	-
Protein, g/dL	5.8 (0.5)	5.7 (2.1)	0.336	NA	-
Ammonium, µmol/L	62 (20–92) **	201 (-)	0.013	NA	-
Peak					
ALT, UI/L	732 (335)	232 (303)	0.005	NA	-
AST, UI/L	811 (223)	163 (386)	0.023	NA	-
GGT, UI/L	1084 (259)	140 (139)	<0.001	NA	-
ALP, UI/L	195 (146)	146 (104)	0.385	NA	-
TB, mg/dL	4.5 (7.7)	0.5 (0.5)	0.002	NA	-
PA, %	64.88 (32.8)	81.3 (8.7)	0.147	NA	-
LDH, UI/L	652 (50)	513 (32)	0.005	NA	-
Cr, mg/dL	1.04 (0.67)	0.82 (0.55)	0.668	NA	-
Albumin, g/dL	3.1 (0.6)	2.7 (1.3)	0.362	NA	-
Protein, g/dL	5.2 (0.8)	6.0 (2.1)	0.245	NA	-
Ammonium, µmol/L	142.5 (75–432) **	-	-	NA	-
Recovery					
ALT, UI/L	77 (106)	58 (98)	0.64	NA	-
AST, UI/L	56 (63)	30 (14)	0.219	NA	-
GGT, UI/L	90 (89)	76 (127)	0.342	NA	-
ALP, UI/L	110 (66)	72 (39)	0.222	NA	-
TB, mg/dL	7.1 (17.1)	1.2 (2.2)	0.242	NA	-
PA, %	84.8 (20.0)	95.3 (8.7)	0.154	NA	-
LDH, UI/L	246 (106.7)	255 (76.7)	0.861	NA	-
Cr, mg/dL	0.84 (0.45)	0.79 (0.30)	0.411	NA	-
Albumin, g/dL	3.6 (0.6)	3.8 (0.5)	0.401	NA	-
Protein, g/dL	6.4 (0.5)	6.2 (2.1)	0.893	NA	-
Ammonium, µmol/L	101 (66–116) **	-	-	NA	-
VPA concentration					
VPA 1, ug/mL	59.4 (28.3)	59.6 (20.6)	0.985	57.9 (20.8)	0.003
Time since onset, days	−38 (−12)	−49 (19)	0.122	−31 (12)	0.437
VPA 2, ug/mL	56.3 (22.2)	58.3 (27.5)	0.92	57.7 (26.5)	0.152
Time since onset, days	−1.17 (3.7)	10 (13)	0.097	9 (9)	0.897
VPA 3, ug/mL	51.0 (16.8)	76.3 (19.4)	0.018	61.6 (16.2)	0.014
Time since onset, days	5 (9.8)	44.8 (57.3)	0.004	34.6 (67.1)	0.090

Values are presented as mean (± standard deviation) unless otherwise noted. (*) Matching variables. (**) sig. *p* = 0.043. ^(**#**)^ Ref. iDOctus v2.3.401 (1) ESGPr23401001, last update 21 March 2020, 21:46. Abbreviations: ALT, alanine aminotransferase; ALP, alkaline phosphatase; AST, aspartate transaminase; BMI, body mass index; CNS, central nervous system; Cr, creatinine; G, grade; GGT, gamma-glutamyl transferase; LDH, lactate dehydrogenase; PA, prothrombin activity; TB, total bilirubin; VPA, valproic acid; SD, standard deviation; NA, not applicable. The patients who developed mild ALT elevation showed more previous drug allergies (6.7% vs. 26.6%, *p* < 0.001), while there was no statistically significant difference between the cases and the healthy control group (6.7% vs. 10%, *p* = 0.3). There were also no statistically significant differences in the number of concomitant drugs in any of the study groups. There was a higher prevalence of alcoholism among the patients who developed DILI compared with those with mild liver injury or no ADR (53.3% vs. 13.3% vs. 3.3%, *p* < 0.01).

**Table 2 jcm-10-01153-t002:** Significant factors and covariates in the multiple logistic regression for drug-induced liver injury.

	Pseudo-OR *	95% BCa CI	*p*-Value
Toxics (alcohol)	7.30	6.17–8.44	<0.001
Co-antiepileptics, *n*	4.33	2.57–6.09	<0.001
Indication (Epilepsy)	2.34	1.94–2.74	0.002
Co-medication, *n*	1.58	1.39–1.77	0.001
Lactate dehydrogenase, IU/L	1.41	1.18–1.64	0.02
Pseudo R^2^ = 0.908			

* Based on 1000 bootstrap samples; Abbreviations: BCa, bias corrected and accelerated; *n*, number; OR, odds ratio.

## Data Availability

The data that support the findings of this study are available from the corresponding author upon reasonable request.

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
