# Peer review of "Valproic Acid-Induced Liver Injury: A Case-Control Study from a Prospective Pharmacovigilance Program in a Tertiary Hospital"

_jcm, 2021, doi:10.3390/jcm10061153_

Round 1

Reviewer 1 Report

Dear authors,

I am happy for the chance of reviewing this paper, and apologies for the delay. This is a clear and well-conducted case-control study, which points to some interesting results compared to the existing literature. It is true that the sample size is not big, however for this study design it is fine. Compared to the Vigibase analysis of ICSRs you quote, it has the advantage of Control Groups as you comment, and it nicely confirms the observation of polytherapy as risk factor. Seizure indication is probably Connected to the polytherapy, and not a Direct risk factor, but you do point out that in the discussion.

I only have suggestions of personal preference, which the authors might or might not implement (not required). 1)I would have liked to see a sentence about which kind of evidence you suggest would be required in future Research to confirm Your findings (you only say 'closely monitored'). 2) That huge picture of the VPA mechanism of action, taken from a previous publication, has too much emphasis I think. It can be made smaller at least. Rather add details about the hypothetical mechanism of liver toxicity. 3) Since genotyping is missing in this study (which might have been Nice seen POLG mutation  as Genetic risk factor) maybe a comment about that would have a place in the discussion - future research.

Author Response

Dear reviewer 1

We would like to thank to the reviewer for his(her) comments and suggestions for the manuscript. We have carefully revised the manuscript accordingly to the reviewer´s suggestions. Below you can see the respond to the comments. The changes in the original manuscript have been made using the track changes mode in MS Word.

I only have suggestions of personal preference, which the authors might not implement (not required), 1)I would have liked to see a sentence about which kind of evidence you suggest would be required in future Research to confirm Your findings (you only say “closely monitored”), 2) That huge picture of the VPA mechanism of action, taken from a previous publication, has too much emphasis I think. It can be made smaller at least, Rather add details about the hypothetical mechanism of liver toxicity. 3)Since genotyping is missing in this study (which might have been Nice seen POLG mutation as Genetic risk factor) maybe a comment about that would have a place in the discussion – future research.

Response:  We really appreciate your comments on the paper. We have made some changes according to your suggestions: 1) Added sentence that future research is needed to confirm theses findings  2) Removed figure of VPA mechanism of action from the introduction  3) Some sentences in the discussion about the possible relevance of genotyping have been added. A figure to explain hypothetical mechanism damage due to liver injury of the VPA has been added.

Regards

Reviewer 2 Report

The authors investigated risk factors for developing liver injury in patients treated with valproic acid (VPA). They found that concomitant intake of alcohol, co-antiepileptics, epilepsy as a cause for treatment with VPA, other co-medications than antiepileptics and increased lactate dehydrogenase activity before starting the treatment were such risk factors.

I have the following remarks and questions:

  1. Introduction: in my view too long. The extensive description (including the figure) reminds on review article, not on an article presenting original data. This should be focused on the problem presented.
  2. Page 4/case definition and entire paper: “The cause and type of hepatitis …”. The term hepatitis is in my view based on a histological assessment. If not all patients were biopsied and cell infiltrates suggesting hepatitis was found (what I doubt), the term liver injury (which can be hepatocellular, cholestatic or mixed) should be used.
  3. Page 5/sample size calculation: I do not know what the sentence “It is assumed that the rate of exposed in the control group will be 0.35-0.4” means. Rate of what?
  4. Page 6/Table 1: “Drug allergy” should be specified
  5. Page 7/Table 1: Alcohol consumption: This should be quantified. On page 11, the authors state that the rate of alcoholism was higher. A high alcohol consumption must not mean alcoholism. The authors should give the amount of alcohol ingested. Interactions: red, yellow and green should be defined. Which were the interacting drugs identified? Also CYP inducers? Which antiepileptics were used as co-tretament?7.
  6. Page 8/Table 1: Nor recovered should be changed to not recovered. Significant numbers for enzyme activities: numbers after the comma should not be given (the error for the determination is in the range of 5 to 10%).
  7. Page 12/par 2: By which mechanisms could alcohol increase the toxicity of VPA? This should be discussed more precisely.
  8. Pages should be numbered correctly.

Author Response

Dear reviewer 2

We would like to thank to the reviewer for his(her) comments and suggestions for the manuscript. We have carefully revised the manuscript accordingly to the reviewer´s suggestions. Below you can see the respond to the comments. The changes in the original manuscript have been made using the track changes mode in MS Word.

Reviewer 2: The authors investigated risk factors for developing liver injury in patients treated with valproic acid (VPA). They found that concomitant intake of alcohol, co-antiepileptics, epilepsy as a cause for treatment with VPA, other co-medications than antiepileptics and increased lactate dehydrogenase activity before starting the treatment were such risk factors. I have the following remarks and questions: 1. Introduction: in my view too long. The extensive description (including the figure) reminds on review article, not an article presenting original data. This should be focus on the problem presented.

Response:  We decided to remove figure from the introduction in order to reduce the extension.

Reviewer 2: 2. Page 4/case definition and entire paper: “The cause and type of hepatitis…”, The term hepatitis is in my view based on a histological assessment. If not all patients were biopsied and cell infiltrates suggesting hepatitis was found (what I doubt), the term liver injury (which can be hepatocellular, cholestatic or mixed) should be used.

Response:  We have replaced the term hepatitis for liver injury in all situations referred to our cases and controls, but we kept the term when theoretically speaking.

Reviewer 2: 3. Page 5/sample size calculation: I do not know what the sentence “It is assumed that the rate of exposed in the control group will be 0,35-0,4”. Rate of what?

Response: We correct the sentence to “A proportion of exposed subjects in the control group has been estimated to be 0.3%”. 

Reviewer 2: 4. Page 6/Table1: “Drug allergy” should be specified.

Response:  Information about specific drugs allergies has been included.

Reviewer 2: 5. Page7/Table1: Alcohol consumption: This should be quantified. On page 11, the authors state that the rate of alcoholism was higher. A high alcohol consumption must not mean alcoholism. The authors should give the amount of alcohol ingested. Interactions: red, yellow and green should be defined. Which were the interacting drugs identified? Also CYP inducers? Which antiepileptics were used as co-treatment?7.

Response:  One of the limitations of a retrospective observational study is the limitation of data source, information on alcohol consumption was obtained from medical records, based on the physician's clinical judgement in what they considered to be an excessive alcohol consumption that could affect patient´s health. We try to clarify this point in Material and Methods section.

The definition of drug interactions has been defined on the basis of “Consejo General de Colegios Oficiales de Farmacéuticos” (General Council of Official Associations of Pharmacists) of Spain. Red is used for severe interactions which should be avoided; yellow to express moderate interactions that need to be aware and treatment modification considered; green for minor interactions that should be known but not imply change of medication.  (This information has been also included in the article)

Table 1 has been updated to attach registration of concomitant antiepileptic medication for each group.

Reviewer 2: 6. Page8/Table 1: Nor recovered should be changed to not recovered. Significant numbers for enzyme activities: numbers after the comma should not be given (the error for determination is in the range of 5 to 10%).

Response:  We have corrected these mistakes on Table 1.

Reviewer 2: 7. Page12/par 2: By which mechanisms could alcohol increase the toxicity of VPA? This should be discussed more precisely.

Response:  We decided to address this issue in more depth in the discussion.

Reviewer 2: 8. Pages should be numbered correctly.

Response:  We have taken into account your comment and we corrected the page numbering. Thank you.

 Regards

Reviewer 3 Report

Information missing in what concerns other signs/symptoms concomitant to altered laboratory values for cases, drug interactions not described (type of drugs, definition for alcohol/tobacco consumption) ; no information about differential diagnosis nor data of infectious diseases, nor DRESS syndrome cases 

Mechanisms of DILI not discussed , focus on laboratory values but not clinical diagnosis hypothesis described (immunoallergic origin, toxic origine by genetic condition, metabolism, etc)

Time to onset for DILI between both groups is clearly different, no explanation is mentioned in the discussion

Numbers in table for cases for medical history are incorrect.

Author Response

Dear reviewer 3

We would like to thank to the reviewer for his(her) comments and suggestions for the manuscript. We have carefully revised the manuscript accordingly to the reviewer´s suggestions. Below you can see the respond to the comments. The changes in the original manuscript have been made using the track changes mode in MS Word.

Reviewer 3: Information missing in what concerns other signs/symptoms concomitant to altered laboratory values for cases, drug interactions not described (type of drugs, definition for alcohol/tobacco consumption); no information about differential diagnosis nor data of infectious diseases, nor DRESS syndrome cases.

Response: Information about drug interactions has been updated including drugs involved. As we already stated, the source of data is limited by medical records, the definition for toxic consumption (alcohol and tobacco) has been added on material and methods, as based on the physician's clinical judgment. No information for infectious diseases was attached as they were all negatives in the cases included, considering VPA related DILI as a diagnosis of exclusion where no other explanation could be found. The causality assessment of drugs was performed using the Roussel Uclaf Causality Assessment Method (RUCAM). A causality score according to RUCAM greater than 5 (6-8 probable; ≥9 highly probable) was considered VPA-induced liver injury. We try to clarify these points in the manuscript.

Reviewer 3: Mechanisms of DILI not discussed, focus on laboratory values but not clinical diagnosis hypothesis described (inmunoallergic origin, toxic origin by genetic condition, metabolism, etc)

Response: We added a discussion the potential mechanism of VPA-induced liver injury.

Reviewer 3: Time of onset for DILI between both groups is clearly different, no explanation is mentioned in the discussion.

Response: According to your comments, a brief explanation has been added to the discussion.

Reviewer 3: Numbers in table for cases for medical history are incorrect.

Response: We reviewed the table of cases based on your observation. We did not find any incorrect information in the medical history. The total number of antecedents does not necessarily match the number of cases, as some patients had more than one medical disorder while others had none.

Regards

Round 2

Reviewer 2 Report

The authors corrected most issues raised in a satisfactory way. I still have some points that should be clarified before acceptance.

  1. Page 13: The hydrogens from alcohol are transferred to NAD+, not to NADP+
  2. Hepatotoxicity of alcohol and VPA: If I understand correctly, the authors propose that the hepatotoxicities of ethanol and VPA are additive and, how they describe it, non-connected. The toxicities are probably additive, but they may be connected. Furthermore, they should be described better, if the authors want to describe them.

Ethanol: The authors state that the toxicity of ethanol is mainly due to increased oxygen consumption due to the production of NADH + H+. This may be true but usually, the production of acetaldehyde (by ADH) and of ROS (by the MEOS) are considered to be responsible for the hepatotoxicity of ethanol. In addition, alcohol (or metabolites) are also toxic for mitochondria and inhibit beta-oxidation, which is also true for VPA and metabolites.

VPA: VPA is mainly glucuronidated in the cytoplasm of the hepatocytes and excreted. About a third is activated to VPA-CoA and transported via the carnitine shuttle into the mitochondria. The carnitine that is used for the shuttle is liberated in the mitochondrial matrix; the transport is no reason for carnitine depletion (by the way, the carnitine concentration is 10 to 100 times higher than the CoA concentration). In the mitochondria, VPA is then beta-oxidized, intermediates are 2-ene-VPA, 3-keto-VPA, propionyl-CoA and pentanoyl-CoA (converted to propionyl-CoA and Ac-CoA). 4-ene-VPA is produced in the ER, can enter mitochondria and be beta-oxidized and inhibits mitochondrial beta-oxidation. In the graph, beta-oxidation appears to be in the cytoplasm (it must be in the mitochondria) and it must be clear that the beta-oxidation products can stem from VPA or 4-ene-VPA, 4-ene-VPA is not a beta-oxidation product. The graph needs therefore to be adapted. In addition: carnitine depletion is possible, but not because of the transport into the mitochondria. Within mitochondria, the CoA-derivatives are reconverted to the carnitine derivatives. In contrast to CoAs, carnitine derivatives can exit the mitochondria and the hepatocyte and be eliminated via the kidney. This is the reason for a (possible) carnitine deficiency. Furthermore, conjugation with GSH is no reason for oxidative stress; conjugation with GSH mitigates oxidative stress. Oxidative stress in mitochondria is due to inhibition of the electron transport chain by VPA/metabolites, leading to the formation of superoxide. Via SOD2 H2O2 is produced, which can be reduced by GSH.

Conclusion: a statement that the toxicities of ethanol and VPA are most probably additive (mitochondrial toxicity, inhibition of beta-oxidation) would be enough for me. If the authors want to describe the toxicity in more detail and provide a graph, this should be correct (see me remarks).

  1. Regarding the interactions, it would be good to know whether CYP inducers were ingested or not. CYP inducers can promote the production of 4-ene-VPA.

Author Response

Dear reviewer

We would like to thank to the reviewer for his(her) comments and suggestions for the manuscript. We have carefully revised the manuscript accordingly to the reviewer´s suggestions. Below you can see the respond to the comments. The changes in the original manuscript have been made using the track changes mode in MS Word.

Reviewer(s)' Comments to Author:
Reviewer 2: The authors corrected most issues raised in a satisfactory way. I still have some points that should be clarified before acceptance.

  1. Page 13: The hydrogens from alcohol are transferred to NAD+, not to NADP+

Response: This mistake has been corrected.

Reviewer 2: 2. Hepatotoxicity of alcohol and VPA: If I understand correctly, the authors propose that the hepatotoxicities of ethanol and VPA are additive and, how they describe it, non-connected. The toxicities are probably additive, but they may be connected. Furthermore, they should be described better, if the authors want to describe them.

Ethanol: The authors state that the toxicity of ethanol is mainly due to increased oxygen consumption due to the production of NADH + H+. This may be true but usually, the production of acetaldehyde (by ADH) and of ROS (by the MEOS) are considered to be responsible for the hepatotoxicity of ethanol. In addition, alcohol (or metabolites) are also toxic for mitochondria and inhibit beta-oxidation, which is also true for VPA and metabolites.

VPA: VPA is mainly glucuronidated in the cytoplasm of the hepatocytes and excreted. About a third is activated to VPA-CoA and transported via the carnitine shuttle into the mitochondria. The carnitine that is used for the shuttle is liberated in the mitochondrial matrix; the transport is no reason for carnitine depletion (by the way, the carnitine concentration is 10 to 100 times higher than the CoA concentration). In the mitochondria, VPA is then beta-oxidized, intermediates are 2-ene-VPA, 3-keto-VPA, propionyl-CoA and pentanoyl-CoA (converted to propionyl-CoA and Ac-CoA). 4-ene-VPA is produced in the ER, can enter mitochondria and be beta-oxidized and inhibits mitochondrial beta-oxidation. In the graph, beta-oxidation appears to be in the cytoplasm (it must be in the mitochondria) and it must be clear that the beta-oxidation products can stem from VPA or 4-ene-VPA, 4-ene-VPA is not a beta-oxidation product. The graph needs therefore to be adapted. In addition: carnitine depletion is possible, but not because of the transport into the mitochondria. Within mitochondria, the CoA-derivatives are reconverted to the carnitine derivatives. In contrast to CoAs, carnitine derivatives can exit the mitochondria and the hepatocyte and be eliminated via the kidney. This is the reason for a (possible) carnitine deficiency. Furthermore, conjugation with GSH is no reason for oxidative stress; conjugation with GSH mitigates oxidative stress. Oxidative stress in mitochondria is due to inhibition of the electron transport chain by VPA/metabolites, leading to the formation of superoxide. Via SOD2 H2O2 is produced, which can be reduced by GSH.

Conclusion: a statement that the toxicities of ethanol and VPA are most probably additive (mitochondrial toxicity, inhibition of beta-oxidation) would be enough for me. If the authors want to describe the toxicity in more detail and provide a graph, this should be correct (see me remarks).

Response: The article´s discussion and bibliography has been updated in order to extend about this topic. Graph has been removed to avoid any possible confusion.

Reviewer 2: 3. Regarding the interactions, it would be good to know whether CYP inducers were ingested or not. CYP inducers can promote the production of 4-ene-VPA.

Response: A brief commentary on which of the drug interactions could affect CYP has been added.

Regards,

Enrique Seco Meseguer

Elena Ramírez
